# LAMOL: LAnguage MOdeling for Lifelong Language Learning

**Fan-Keng Sun**[*][†]
MIT
Cambridge, MA, USA
fankeng@mit.edu

**Cheng-Hao Ho**[*]
National Taiwan University
Taipei, Taiwan
jojotenya@gmail.com

**Hung-Yi Lee**
National Taiwan University
Taipei, Taiwan
hungyilee@ntu.edu.tw

## Abstract

Most research on lifelong learning applies to images or games, but not language. We present LAMOL, a simple yet effective method for lifelong language learning (LLL) based on language modeling. LAMOL replays pseudo-samples of previous tasks while requiring no extra memory or model capacity. Specifically, LAMOL is a language model that simultaneously learns to solve the tasks and generate training samples. When the model is trained for a new task, it generates pseudo-samples of previous tasks for training alongside data for the new task. The results show that LAMOL prevents catastrophic forgetting without any sign of intransigence and can perform five very different language tasks sequentially with only one model. Overall, LAMOL outperforms previous methods by a considerable margin and is only 2–3% worse than multitasking, which is usually considered the LLL upper bound. The source code is available at https://github.com/jojotenya/LAMOL.

## 1 Introduction

The current dominant paradigm for machine learning is to run an algorithm on a given dataset to produce a trained model specifically for a particular purpose; this is *isolated learning* (Chen & Liu, 2016, p. 150). In isolated learning, the model is unable to retain and accumulate the knowledge it has learned before. When a stream of tasks are joined to be trained sequentially, isolated learning faces *catastrophic forgetting* (McCloskey & Cohen, 1989) due to a non-stationary data distribution that biases the model (left figure of Figure 1). In contrast, lifelong learning is designed to address a stream of tasks by accumulating interconnected knowledge between learned tasks and retaining the performance of those tasks. A human easily achieves lifelong learning, but this is nontrivial for a machine; thus lifelong learning is a vital step toward artificial general intelligence.

In this paper, we focus on lifelong language learning, where a machine achieves lifelong learning on a stream of natural language processing (NLP) tasks. To the best of our knowledge, lifelong language learning has been studied in only a few instances; for sentiment analysis (Chen et al., 2015b; Xia et al., 2017), conversational agents (Lee, 2017), word representation learning (Xu et al., 2018), sentence representation learning (Liu et al., 2019), text classification, and question answering (d'Autume et al., 2019). However, in all previous work, the tasks in the stream are essentially the same task but in different domains. To achieve lifelong language learning on fundamentally different tasks, we propose LAMOL — LAnguage MOdeling for Lifelong language learning.

It has been shown that many NLP tasks can be considered question answering (QA) (Bryan McCann & Socher, 2018). Therefore, we address multiple NLP tasks with a single model by training a language model (LM) that generates an answer based on the context and the question. Treating QA as language modeling is beneficial because the LM can be pre-trained on a large number of sentences without any labeling (Radford et al., 2019); however, this does not directly solve the problem of LLL. If we train an LM on a stream of tasks, catastrophic forgetting still occurs. However, as an LM is intrinsically a text generator, we can use it to answer questions while generating pseudo-samples of

---

[*]Equal contribution.
[†]Work done while at National Taiwan University.

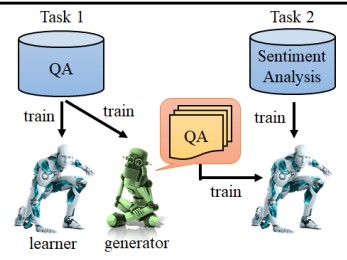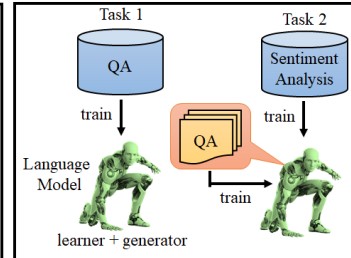

Figure 1: **Left**: After learning Task 2, the learner has already forgotten how to solve Task 1. This is "catastrophic forgetting". **Middle**: The basic idea of the data-based LLL approach. A generator is learned to generate examples it has seen before. Using the generator, the learner also learns from examples from the previous task to prevent it from forgetting. **Right**: A language model that simultaneously takes on the roles of learner and generator.

the previous task to be replayed later. LAMOL is inspired by the data-based approach for LLL in which a generator learns to generate samples in previous tasks (middle of Figure 1) (Hanul Shin & Kim, 2017; Kemker & Kanan, 2017). In contrast to previous approaches, LAMOL needs no extra generator (right of Figure 1). LAMOL is also similar to multitask training, but the model itself generates data from previous tasks instead of using real data.

Our main contributions in this paper are:

- We present LAMOL, a simple yet effective method for LLL. Our method has the advantages of no requirements in terms of extra memory or model capacity. We also do not need to know how many tasks to train in advance and can always train on additional tasks when needed.

- Experimental results show that our methods outperform baselines and other state-of-the-art methods by a considerable margin and approaches the multitasking upper bound within 2–3%.

- Furthermore, we propose adding task-specific tokens during pseudo-sample generation to evenly split the generated samples among all previous tasks. This extension stabilizes LLL and is particularly useful when training on a large number of tasks.

- We analyze how different amounts of pseudo-samples affect the final performance of LAMOL, considering results both with and without the task-specific tokens.

- We open-source our code to facilitate further LLL research.

## 2 RELATED WORK

Lifelong learning research is based on regularization, architecture, or data. Here is a brief survey of works in these three categories.

### 2.1 REGULARIZATION-BASED METHODS

In this approach, a constraint, i.e., a regularization term, is added to minimize deviation from trained weights while updating the weights in a new task. Most regularization based methods estimate the importance of each parameter and add the importance as a constraint to the loss function. Elastic weight consolidation (EWC) (Kirkpatrick et al., 2017) calculates a Fisher information matrix to estimate the sensitivity of parameters as importance. Online EWC (Schwarz et al., 2018) is a transformed version of EWC. Instead of tracking the importance of parameters for each task, online EWC simply accumulates the importance of the stream of tasks. Synaptic intelligence (SI) (Zenke et al., 2017) assigns importance to each parameter according to its contribution to the change in the total loss. Memory aware synapses (MAS) (Aljundi et al., 2018) estimate importance via the gradients of the model outputs. In contrast to estimating the importance of weights, incremental moment matching (IMM) (Lee et al., 2017) matches the moment of weights between different tasks.

## 2.2 ARCHITECTURE-BASED METHODS

For this category, the main idea is to assign a dedicated capacity inside a model for each task. After completing a task, the weights are frozen and may not be changed thereafter. Some methods allow models to expand, whereas some fix the size but must allocate capacity for tasks at the beginning. Progressive neural networks (Rusu et al., 2016) utilize one column of the neural network per task. Once a new task is trained, progressive neural networks augment a new column of the neural network for the task while freezing the past trained columns. Columns that have been frozen are not allowed to change but are connected to the new column to transfer knowledge from old tasks. Towards Training Recurrent Neural Networks for Lifelong Learning (Sodhani et al., 2018) unifies Gradient episodic memory (Lopez-Paz et al., 2017) and Net2Net (Chen et al., 2015a). Using the curriculum-based setting, the model learns the tasks in easy-to-hard order. The model alleviates the forgetting problem by GEM method, and if it fails to learn the current task and has not been expanded yet, the model will expand to a larger model by the Net2Net approach.

PathNet (Fernando et al., 2017) reuses subsets of a neural network to transfer knowledge between tasks. Unlike progressive neural networks, PathNet does not allow the model to expand. Instead, it builds a huge fixed-size model composed of a neural network and paths between different layers of the neural networks. While training a task, it selects the best combination of neural networks and paths for that particular task. Similar to progressive neural networks, selected parts are fixed to allow only inference and not training. Inspired by network pruning, PackNet (Mallya & Lazebnik, 2018) prunes and re-trains the network iteratively to pack numerous tasks into a single huge model.

This category has some drawbacks. When resources are limited, model expansion is prohibited. Also, some architecture-based methods require the number of tasks in advance to allocate the capacity for the tasks, which greatly reduces their practicality.

## 2.3 DATA-BASED METHODS

This method restricts weights through the data distribution of old tasks. One data-based approach keeps a small amount of real samples from old tasks, and the other distills the knowledge from old data and imagines pseudo-data of old tasks later on. While training a new task, the data or pseudo-data is used to prevent weights from greatly deviating from the previous status.

Gradient episodic memory (GEM) (Lopez-Paz et al., 2017) preserves a subset of real samples from previous tasks. Utilizing these real samples during optimization helps somewhat to constrain parameter gradients. Averaged-GEM (A-GEM) (Chaudhry et al., 2018) is a more efficient version of GEM which achieves the same or even better performance than the original GEM. Learning without forgetting (Li & Hoiem, 2017) minimizes the alteration of shared parameters by recording the outputs from old task modules on data from the new task before updating. Hanul Shin & Kim (2017) and Kemker & Kanan (2017) encode data from old tasks into a generative model system. The latter imitates the dual-memory system of the human brain, in that the model automatically decides which memory should be consolidated. Both methods replay pseudo-data of previous tasks using the generative model during training.

d'Autume et al. (2019) investigates the performance of the episodic memory system on NLP problems. It distills the knowledge of previous tasks into episodic memory and replays it afterward. This work evaluates the method on two streams of tasks: question answering and text classification.

## 3 LAMOL

A pre-trained LM can generate a coherent sequence of text given a context. Thus, we propose LAMOL, a method of training a single LM that learns not only to answer the question given the context but also to generate the context, the question, and the answer given a generation token. That is, in LAMOL, a model plays the role of both LM and QA model. Hence, answering questions and generating pseudo-old samples can both be done by a single model. During LLL, these pseudo-old samples are trained with new samples from new tasks to help mitigate catastrophic forgetting.

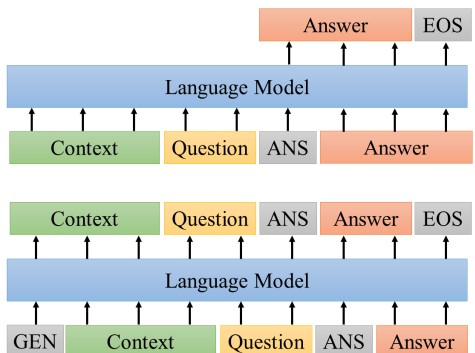

Figure 2: **Upper**: LM learns to answer question given context. **Lower**: LM learns to generate training samples given generation token GEN.

### 3.1 DATA FORMATTING

Inspired by the protocol used by decaNLP (Bryan McCann & Socher, 2018), samples from the datasets we used are framed into a SQuAD-like scheme, which consists of context, question, and answer. Although the LM is simultaneously a QA model, the data format depends on the training objective. When training as a QA model, the LM learns to decode the answer after reading the context and question. On the other hand, when training as an LM, the LM learns to decode all three parts given a generation token.

In addition to context, question, and answer, we add three special tokens:

**ANS** Inserted between question and answer. As the context and question are known during inference, decoding starts after inputting ANS.

**EOS** The last token of every example. Decoding stops when EOS is encountered.

**GEN** The first token during pseudo-sample generation. Decoding starts after inputting GEN.

The data formats for QA and LM training are shown in Figure 2.

### 3.2 TRAINING

Assume a stream of tasks $\{T_1, T_2, \dots\}$, where the number of tasks may be unknown. Directly training the LM on these tasks sequentially results in catastrophic forgetting. Thus, before beginning training on a new task $T_i, i > 1$, the model first generates pseudo samples $T_i^{'}$ by top-$k$ sampling that represent the data distribution of previous tasks $T_1, \dots, T_{i-1}$. Then, the LM trains on the mixture of $T_i$ and $T_i^{'}$. To balance the ratio between $|T_i|$ and $|T_i^{'}|$, the LM generates $\gamma|T_i|$ pseudo samples, where $|T_i|$ denotes the number of samples in task $T_i$ and $\gamma$ is the sampling ratio. If the generated sample does not have exactly one ANS in it, then the sample is discarded. This happens in only 0.5%-1% of generated samples.

During training, each sample is formatted into both the QA format and the LM format. Then, in the same optimization step, both formats are fed into the LM to minimize the QA loss $L_{QA}$ and LM loss $L_{LM}$ together. Overall, the loss is $L = L_{QA} + \lambda L_{LM}$, where $\lambda$ is the weight of the LM loss.

### 3.3 TASK-SPECIFIC TOKENS

Using the same GEN token for all tasks is problematic when training for many tasks because the portion of old tasks decreases exponentially in theory. For instance, if $\gamma = 0.01$, then the portion of the first task when training the second task is about 1%, but is only about 0.01% when training the third task. This issue is definitely harmful to LLL. To mitigate this, we can choose to replace the GEN token with a task-specific token for each task to inform the model to generate pseudo-samples belonging to the specific task. Under this setup, all previous tasks have the same share of the $\gamma|T_i|$ generated pseudo samples. That is, when beginning training for the $i$-th task $T_i$, we generate $\frac{\gamma}{i-1}|T_i|$

| Task | Dataset | # Train | # Test | Metric |
|------|---------|---------|--------|--------|
| Question answering | SQuAD | 87599 | 10570 | nF1 |
| Semantic parsing | WikiSQL | 56355 | 15878 | lfEM |
| Sentiment analysis | SST | 6920 | 1821 | EM |
| Semantic role labeling | QA-SRL | 6414 | 2201 | nF1 |
| Goal-oriented dialogue | WOZ | 2536 | 1646 | dsEM |
| Text classification | AGNews Amazon DBPedia Yahoo Yelp | 115000 | 7600 | EM |

Table 1: Summary of tasks, datasets, dataset sizes, and their corresponding metrics. As this work uses no development set, only the training and test datasets are shown. nF1 is the normalized version of the F1 score; EM represents an exact match between texts: for text classification, this amounts to accuracy; for WOZ, it is equivalent to dfEM (turn-based dialogue state exact match); for WikiSQL, it is equivalent to lfEM (exact match of logical forms).

| | SQuAD | WikiSQL | SST | SRL | WOZ | AGNews | Amazon | DBPedia | Yahoo | Yelp |
|---|-------|---------|-----|-----|-----|--------|--------|---------|-------|------|
| GPT-2 score | 72.3 | 70.7 | **90.9** | 70.4 | **84.9** | **94.6** | **62.3** | **99.1** | **73.9** | **67.7** |
| Other scores | **75.5** | **72.6** | 88.1 | **75.2** | 84.4 | 93.8 | 60.1 | 30.5 | 68.6 | 50.7 |

Table 2: Comparison of GPT-2 and other methods on single task scores. Other scores are retrieved from Bryan McCann & Socher (2018) or d'Autume et al. (2019). Better performance in boldface.

for the previous $i - 1$ tasks. Note that as each task uses a specific token, the vocabulary size and the embedding weight of the LM increase slightly as more tasks are trained.

## 4 EXPERIMENT SETUP

### 4.1 TASKS, DATASETS, AND METRICS

We collect five disparate tasks mentioned in decaNLP (Bryan McCann & Socher, 2018): question answering, semantic parsing, sentiment analysis, semantic role labeling, and goal-oriented dialogue, with a dataset for each task.

Furthermore, to compare our method with d'Autume et al. (2019), we conducted experiments on four text classification tasks: news classification, sentiment analysis, Wikipedia article classification, and question-and-answer categorization with five datasets. We use the procedure from d'Autume et al. (2019) to produce equal-sized datasets.

We do not train on all datasets from both papers due to a lack of computational resources. For each task, there is a corresponding evaluation metric. Table 1 contains a summary of tasks, datasets, and metrics. Additional details are provided in Appendix A. Note that the score of any metric lies between 0 and 100%.

### 4.2 METHODS TO BE COMPARED

All methods use the smallest pre-trained GPT-2 model (Radford et al., 2019)[1] as the LM. Each task is trained for nine epochs; greedy decoding is applied during inference.

- **LAMOL** In all experiments, $k = 20$ in top-$k$ sampling and $\lambda = 0.25$ for weight of the LM loss are set. $\text{LAMOL}_{\text{GEN}}^{\gamma}$ denotes LAMOL with a sampling ratio of $\gamma$, and the same GEN token is used for all tasks. If the task-specific tokens are used, GEN is replaced by TASK.
- **Keep real data** Pseudo-samples are replaced by real samples from previous tasks. The quantity of real samples is equally split between previous tasks. This approach can be considered the upper bound of LAMOL. We denote it as $\text{LAMOL}_{\text{REAL}}^{\gamma}$.

---

[1]https://github.com/huggingface/pytorch-transformers

| Methods | SST SRL WOZ | SST WOZ SRL | SRL SST WOZ | SRL WOZ SST | WOZ SST SRL | WOZ SRL SST | Average | Std |
|---|---|---|---|---|---|---|---|---|
| Fine-tuned | 50.2 | 24.7 | 62.9 | 31.3 | 32.8 | 33.9 | 39.3 | 12 |
| EWC | 50.6 | 48.4 | 64.7 | 35.5 | 43.9 | 39.0 | 47.0 | 8.7 |
| MAS | 36.5 | 45.3 | 56.6 | 31.0 | 49.7 | 30.8 | 41.6 | 8.9 |
| GEM | 50.4 | 29.8 | 63.3 | 32.6 | 44.1 | 36.3 | 42.8 | 11 |
| $\text{LAMOL}_{\text{GEN}}^{0}$ | 46.5 | 36.6 | 56.6 | 38.6 | 44.9 | 45.2 | 44.8 | 6.0 |
| $\text{LAMOL}_{\text{GEN}}^{0.05}$ | 79.6 | 78.9 | 73.1 | 73.7 | 68.6 | 75.7 | 74.9 | 3.4 |
| $\text{LAMOL}_{\text{GEN}}^{0.2}$ | 80.0 | 80.7 | 79.6 | 78.7 | 78.4 | 80.5 | **79.7** | 0.8 |
| $\text{LAMOL}_{\text{TASK}}^{0}$ | 41.0 | 33.5 | 50.1 | 41.9 | 49.3 | 41.5 | 42.9 | 5.2 |
| $\text{LAMOL}_{\text{TASK}}^{0.05}$ | 77.3 | 76.9 | 78.1 | 74.7 | 73.4 | 75.8 | 76.0 | 1.5 |
| $\text{LAMOL}_{\text{TASK}}^{0.2}$ | 79.4 | 79.9 | 80.1 | 78.7 | 79.8 | 79.0 | 79.5 | **0.5** |
| $\text{LAMOL}_{\text{REAL}}^{0.05}$ | 81.0 | 78.9 | 80.1 | 80.9 | 77.7 | 78.0 | 79.4 | 1.2 |
| $\text{LAMOL}_{\text{REAL}}^{0.02}$ | 81.8 | 80.6 | 81.6 | 81.2 | 80.4 | 80.5 | 81.0 | 0.5 |
| Multitasked | | | | 81.5 | | | | |

Table 3: Summary of averaged metric scores for different methods under permuted task orders using models at last epoch of last task. The Average and Std columns respectively are the average and standard deviation of the averaged scores for each row of the methods. Multitasked learning as an upper bound is shown at the bottom.

| Fine-tuned | MAS | $\text{LAMOL}_{\text{GEN}}^{0.05}$ | $\text{LAMOL}_{\text{GEN}}^{0.2}$ | $\text{LAMOL}_{\text{TASK}}^{0.05}$ | $\text{LAMOL}_{\text{TASK}}^{0.2}$ | $\text{LAMOL}_{\text{REAL}}^{0.05}$ | $\text{LAMOL}_{\text{REAL}}^{0.2}$ | Multitasked |
|---|---|---|---|---|---|---|---|---|
| 51.5 | 49.5 | 69.6 | 73.1 | 71.5 | **74.3** || 74.5 | 76.0 | 76.6 |

Table 4: Summary of averaged score on five tasks. The scores are reported as the averaged score over all tasks of the models after training on every task. The rightmost three columns – LAMOL with $\gamma = 0.05$ and $\gamma = 0.2$ of real samples from previous tasks and Multitasked – are upper bounds for comparison. Best performance in boldface.

- **Fine-tune** The model is directly fine-tuned on the stream of tasks, one after another.

- **Multitask learning** All tasks are trained simultaneously. Multitask learning is often seen as an upper bound of lifelong learning. In addition, it is also used to determine whether forgetting is caused by a lack of model capacity.

- **Regularization-based methods** Online EWC (Schwarz et al., 2018) and MAS (Aljundi et al., 2018) are compared. They are chosen because they are more computationally efficient than SI (Zenke et al., 2017) and more memory efficient than IMM (Lee et al., 2017). Additionally, experiments such as Elhoseiny et al. (2018) show that MAS has better performance overall.

- **Gradient Episodic Memory (GEM)** When training each task, we randomly sample data from previous task with the amount equivalent to 5% of the current task size into the memory. In each optimization step, the GEM (Lopez-Paz et al., 2017) approach retrieves all the data in the memory to calculate the gradients for the previous tasks.

- **Improved memory-based parameter adaptation (MBPA++)** Sparse experience replay and local adaptation for LLL as proposed in d'Autume et al. (2019). We also re-implement the paper and report better scores using different hyperparameters.

# 5 EXPERIMENTAL RESULTS

## 5.1 SINGLE TASK

To establish a reference on the capability of the GPT-2 model on every dataset, we trained the model on each dataset independently. The results are shown in Table 2. We observe that the performance of the GPT-2 model is actually quite good, even beating the BERT-based model (d'Autume et al., 2019) on text classification datasets by a large margin. Thus, the GPT-2 model has the potential for superior LLL performance, as long as we can prevent catastrophic forgetting.

## 5.2 SST, QA-SRL, AND WOZ TASKS

For an initial understanding of the performance on all of the methods and the effect of task order, we first conducted a small-scale experiment on three small datasets: SST, QA-SRL, and WOZ. We trained all but the the multitasked method on all six permutations of the task order. The final score for each order was obtained by evaluating the model at the conclusion of the training process. The results are shown in Table 3; we make several observations. Note that LAMOL with $\gamma = 0$ is not the same as Fine-tuned, as the LM loss is still optimized.

- Fine-tuned, EWC, MAS, and LAMOL with $\gamma = 0$ show similar performance and are much worse than LAMOL with $\gamma > 0$.
- LAMOL$_{\text{GEN}}^{0.2}$, our best performing method, is only 1.8 percent away from Multitasked, which implies almost no forgetting during LLL.
- The order of the tasks is crucial to the performance. For instance, the WOZ score drops significantly after training other tasks. Thus, if WOZ is not the last task, the performance is usually noticeably worse.
- When using LAMOL, the performance of old tasks maintains almost the same level throughout the training process. When the sampling ratio $\gamma$ is increased, the performance also increases, especially when increased from 0 to 0.05.
- When $\gamma = 0$, adding task-specific tokens harms performance, because the model must fit additional special tokens that are useless. Adding task-specific tokens is also not helpful if $\gamma = 0.2$. We believe that 0.2 is enough for three tasks; thus task-specific tokens are redundant. However, when $\gamma = 0.05$, task-specific tokens are beneficial because the tokens are needed to help retain a substantial presence of the first task when training the third task.
- We see that a better LLL method usually has a smaller standard deviation, which implies that it is effected less by task order. Adding task-specific tokens also has a stabilizing effect.

The complete forgetting progress is illustrated in Appendix B. Clearly, Fine-tuned, EWC, MAS, LAMOL$_{\text{GEN}}^{0}$, and LAMOL$_{\text{TASK}}^{0}$ reveal similar patterns. However, the proposed LAMOL with $\gamma > 0$ displays the ability to retain its learned knowledge. In the case of WOZ $\rightarrow$ SRL $\rightarrow$ SST, the WOZ score even increases after training the third task using LAMOL with $\gamma = 0.2$.

## 5.3 FIVE DECANLP TASKS

Here, we train the following five tasks sequentially: SQuAD, WikiSQL, SST, QA-SRL, and WOZ. Given the limited computing resources, we explore only one task order: from large to small tasks, according to the number of training samples.

As shown in Table 4, LAMOL outperforms all baselines by a large margin and on average approaches within 2–3% of the multitasked upper bound. Also, as expected, the performance of LAMOL improves as the sampling ratio $\gamma$ increases and task-specific tokens are used.

There is also a gap between our method and the method of keeping real samples. As shown in the table, using real samples is much more sample-efficient, as 5% of real samples beats 20% of pseudo-samples. This may be due to the less-than-ideal quality of the pseudo-data. The longer the paragraphs are, the harder it is for the model to create high-quality samples. After observing the samples generated when using task-specific tokens, we discover some "chaos". That is, some examples generated by the model do not exactly correspond to the task-specific token. This implies that the task-specific tokens are sometimes too weak to constrain the model; thus their influence is overshadowed by other tokens. We believe that solving this problem will bring the performance when using task-specific tokens closer to using real samples; however, we leave this as future work.

Figure 3 illustrates the test scores of each method on each task throughout the training. We clearly see that when using LAMOL, the model remembers nearly perfectly.

We make several observations:

- When training SQuAD, QA-SRL has not been trained yet, but the score of QA-SRL is already around 40. Also, when training QA-SRL, the SQuAD score revives if the model

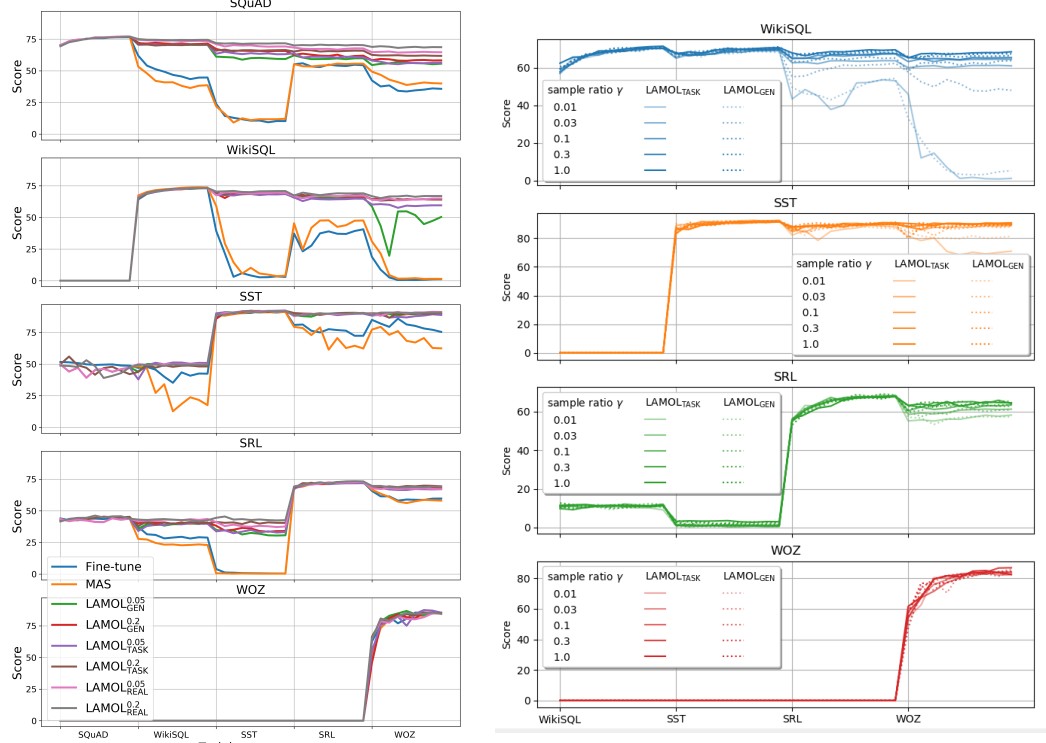

Figure 3: Training progress of five tasks. The graph records the performance of the model at each epoch of each task.

Figure 4: Performance after each epoch under five different sampling ratios, with or without task specific-specific tokens.

has forgotten SQuAD. These two facts imply that SQuAD and SRL are similar tasks, such that the model is capable of transferring knowledge from one to the other.

- If forward transfer exists, replaying pseudo-data also retains the forward transfer. That is, the QA-SRL score does not drop after training on WikiSQL and SST when LAMOL is used but drops significantly for other methods.

- The transferability between SQuAD and QA-SRL is expected. On the other hand, the transferability between WikiSQL and QA-SRL is quite surprising; the WikiSQL score improves considerably when training on QA-SRL for Fine-tuned and MAS after WikiSQL is forgotten during SST training.

## 5.4 TEXT CLASSIFICATION TASKS

We compared the proposed method against the state-of-the-art MBPA++ proposed in d'Autume et al. (2019), both by citing their original numbers and also by reproducing their methods. We chose text classification as opposed to QA because we believe that LM has more of a disadvantage in text classification than in QA. We compared with $\text{LAMOL}_{\text{TASK}}^{0.2}$ due to its good performance and stability. Following their paper and testing our model on the same four kinds of task orders, the results are shown in Table 5.

Our implementation results in much higher scores than the original ones. However, the proposed $\text{LAMOL}_{\text{TASK}}^{0.2}$ still outperforms our implementation of MBPA++.

## 5.5 INFLUENCE OF SAMPLING RATIO $\gamma$

As the value of $\gamma$ determines the performance of LLL, we conducted a medium-scale experiment to understand the influence of $\gamma$ with and without task-specific tokens. In this experiment we used

| Order | MBPA++ | MBPA++ (our impl.) | LAMOL$_{\text{TASK}}^{0.2}$ |
|---|---|---|---|
| i | 70.8 | 74.1 | **76.7** |
| ii | 70.9 | 74.9 | **77.2** |
| iii | 70.2 | 73.1 | **76.1** |
| iv | 70.7 | 74.9 | **76.1** |
| Average | 70.7 | 74.2 | **76.5** |

Table 5: Summary of results on text classification tasks using averaged EM score (equivalent to averaged accuracy in d'Autume et al. (2019)) of models at last epoch of last task. The four orders mirror those in d'Autume et al. (2019). For MBPA++ (out impl.) and LAMOL$_{\text{TASK}}^{0.2}$, the results are averaged over two runs. The $p$-value of pairted $t$-test between eight numbers of MBPA++ (our impl.) and LAMOL$_{\text{TASK}}^{0.2}$ is smaller than 1%, which shows that there is significant difference. Our implementation of MBPA++ is available at `https://github.com/Daikon-Sun/EM-in-LLL`.

WikiSQL (blue color), SST (orange), QA-SRL (green), and WOZ (red), in that training order. The results are shown in Figure 4.

Unsurprisingly, the less generation done by the model, the more likely the vanishing distribution in Section 3 occurs: the model forgets how to generate previous tasks, as the ratio of previous tasks in the total dataset decreases exponentially over time. Models using task-specific tokens mitigate this somewhat, as demonstrated in the first subgraph where the performance of LAMOL$_{\text{TASK}}^{0.03}$ is much better than that of LAMOL$_{\text{GEN}}^{0.03}$.

In addition, the more samples the model generates, the better the overall performance of the model. However, this performance gain disappears when the sampling ratio $\gamma$ is around 0.1 to 0.3.

## 6 CONCLUSION

We propose LAMOL, a simple yet effective method for LLL based on language modeling. A single LM achieves LLL without additional model components and without keeping old examples. Moreover, any pre-trained LM can be used to leverage a large amount of unlabeled text to improve LLL. Finally, more tasks can be added whenever needed.

## ACKNOWLEDGEMENT

This work was supported by the Ministry of Science and Technology of Taiwan.

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

## A  TASKS, DATASET, AND METRICS

Five tasks and their corresponding datasets from decaNLP (Bryan McCann & Socher, 2018):

- **Question Answering – Stanford Question Answering Dataset (SQuAD)** (Rajpurkar et al., 2016): This dataset consists of context, questions, and answers. The context is paragraphs from English Wikipedia, and the answers are spans from its corresponding question paragraphs. For evaluation, we use the normalized F1 score (nF1), which strips out articles and punctuation as in Bryan McCann & Socher (2018). Test datasets in this task are hidden from the host so that users must upload models to their platform to generate the test results; due to this inconvenience and our many models, we elected to use the development set to test the metric. Note that we do not use the development set in the training process. The size of the training set is 87,599 while that of the development set is 10,570.

- **Semantic Parsing – WikiSQL** (Zhong et al., 2017): In this task, normal sentences are translated into SQL-structured SQL queries. WikiSQL provides logical forms along with natural language utterances. The exact match of the logical forms (lfEM) is used to evaluate the performance. The model outputs are required to be matched the SQL format. Otherwise, its won't get any score. The size of the training set is 56,355; that of the test set is 15,878.

- **Sentiment Analysis – Stanford Sentiment Treebank (SST, binary version)** (Radford et al., 2017): This dataset consists of movie reviews with its answers, including positive and negative binary options. The exact match score is used as the metric. The size of the training set is 6,920; that of the test set is 1,821.

- **Semantic Role Labeling – QA-SRL** (He et al., 2017): QA-SRL is a question answering form of the SRL task. The normalized F1 (nF1) score is used. The size of the training set is 6,414; that of the test set is 2,201.

- **Goal-Oriented Dialogue – English Wizard of Oz (WOZ)** (Wen et al., 2016): WOZ is a restaurant reservation task that provides a predefined ontology of a series of information for helping an agent to make reservations for customers. To keep track of the dialogue state, turn-based dialogue state EM (dsEM), which requires the model outputs exactly follow the characters' conversation order, is used for judgment. The size of the training set is 2,536; that of the test set is 1,646.

Four text classification tasks and five datasets from MBPA++ (dAutume et al. 2019):

- **News Classification – AGNews:** News articles to be classified into 4 classes.
- **Sentiment Analysis – Yelp and Amazon:** Customer reviews and ratings on Yelp and Amazon. Both datasets include 5 classes.
- **Wikipedia Article Classification – DBPedia:** Articles and their corresponding categories on Wikipedia, including 14 classes.
- **Questions and Answers Categorization – Yahoo:** Questions and answers on the Yahoo! platform, including 10 classes.

The dataset collected by Xiang Zhang (2015) is available at http://goo.gl/JyCnZq. Given the unbalanced dataset sizes, we randomly sample 115,000 training examples and 7,600 test examples from all the datasets per d'Autume et al. (2019). All the tasks use exact match accuracy as the evaluation metric.

# B  OVERVIEW OF THE FORGETTING PROGRESS FOR THREE TASKS

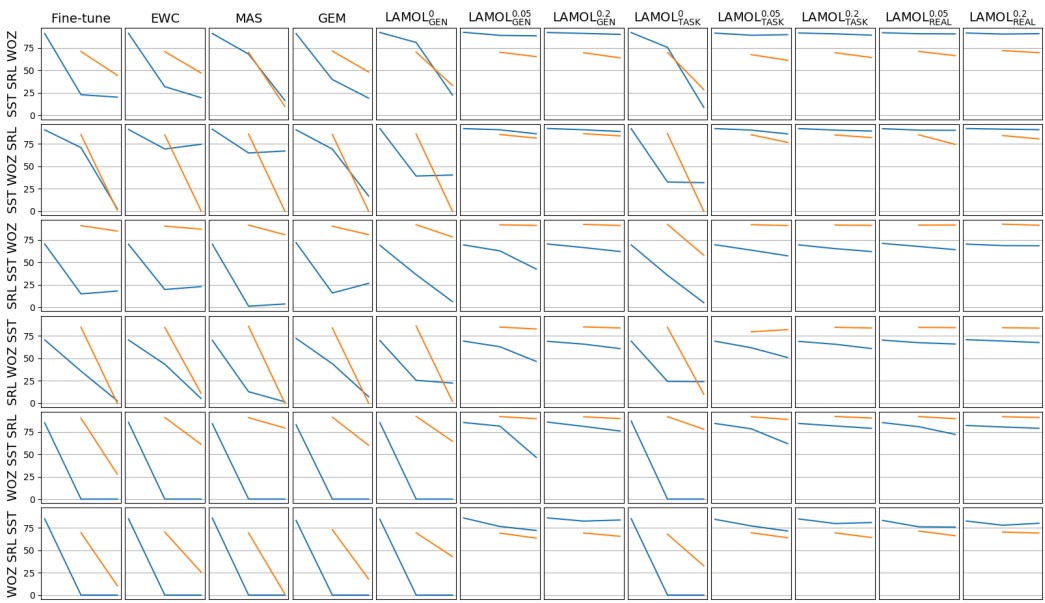

Figure 5: Overview of the forgetting progress for different methods and permuted orders. The blue line indicate the scores of the first task after training each task. The orange line corresponds to that of the second task.

## C    REVERSE ORDER OF FIVE DECANLP TASKS

| Fine-tuned | MAS | LAMOL$_{GEN}^{0.05}$ | LAMOL$_{GEN}^{0.2}$ | LAMOL$_{TASK}^{0.05}$ | LAMOL$_{TASK}^{0.2}$ | ‖ | LAMOL$_{REAL}^{0.05}$ | LAMOL$_{REAL}^{0.2}$ | Multitasked |
|---|---|---|---|---|---|---|---|---|---|
| 45.4 | 44.7 | 63.2 | 73.0 | 75.3 | **76.9** | ‖ | 75.9 | 78.2 | 76.6 |

Table 6: Summary of averaged score on reversed five tasks. The scores are reported as the averaged score over all tasks of the models after training on every task. The rightmost three columns – LAMOL with $\gamma = 0.05$ and $\gamma = 0.2$ of real samples from previous tasks. Best performance in boldface.

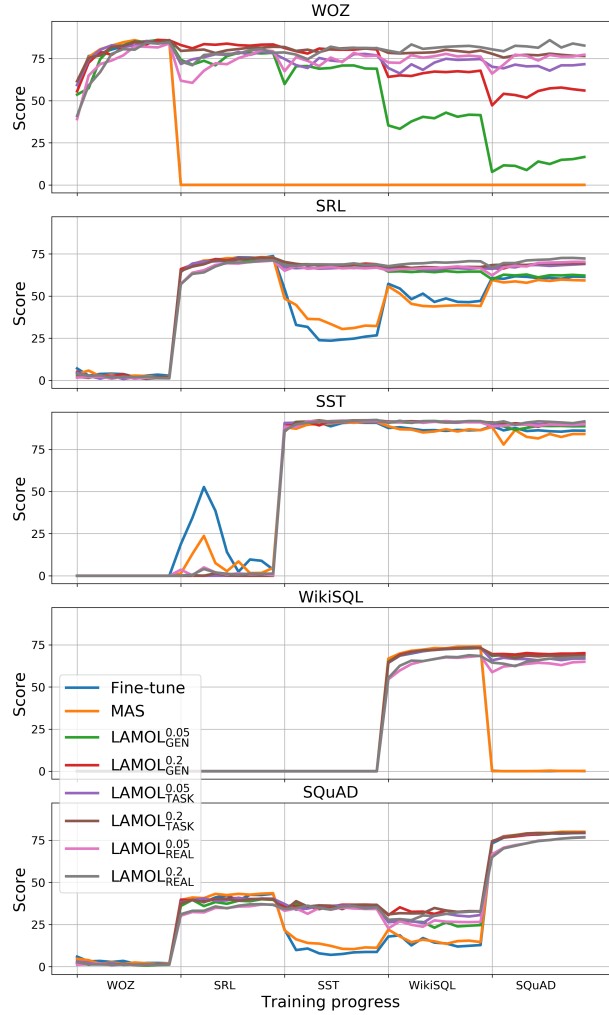

Figure 6: Training progress of the five tasks with reverse order, i.e. small to large. The graph records the performance of the model at each epoch of each tasks. The order of tasks in the progress follows: WOZ, QA-SRL, SST, WikiSQL, and then SQuAD. From the figure, we can clearly see that Fine-tune and MAS completely forget WOZ and WikiSQL.

# D   GENERATED EXAMPLES

---

**Normal Examples**

*‿‿squad1‿‿The United States has been accused of a wide ranging war in Afghanistan since 9 / 11. During the campaign, US forces in Afghanistan were involved in an extensive air campaign. At least 1,600 American servicemen and women were killed, while more than 1,600 civilians were injured. After the US-led invasion of Afghanistan on 12 / 11 / 2001, an estimated 10,000 American soldiers were killed in combat.What were the targets included in the conflict? ‿‿ans‿‿Afghanistan*

*‿‿squad1‿‿In 1849, the French army was forced to withdraw, and the French were finally expelled, although it was not until late November that the French recaptured most of their territories. French troops then reached Egypt. On 21 January 1852 (the year after he left), in Cairo, they captured Tripoli, Benghazi, Benghazi, and the eastern part of Libya. After Gaddafi's return to office, he established the Gaddafi regime. On 13 February 1856, the Gaddafi family relocated to Egypt. On 13 May 1857, the army was forced to withdraw from Libya, and the army returned to Benghazi.On whom did Gaddafi's army return to Benghazi?‿‿ans‿‿Gaddafi's family*

*‿‿wikisql‿‿the table has columns rank, athlete, sport, nation, gold, silver and key words max, min, count, sum, avg, =, >, <, op, select, where, and, col, table, caption, page, section, op, cond, question, agg, aggops, condops - - what is the nation when they have the rank 10 and the gold is 6?what is the translation from english to sql?‿‿ans‿‿select nation from table where rank = 10 and gold = 6*

*‿‿wikisql‿‿the table has columns home team, home team score, away team, away team score, venue, crowd, date and key words max, min, count, sum, avg, =, >, <, op, select, where, and, col, table, caption, page, section, op, cond, question, agg, aggops, condops - - what is the home team's score when they played at richfield?what is the translation from english to sql?‿‿ans‿‿select home team score from table where venue = richfield*

*‿‿sst‿‿it's a movie that's not worth a look.is this review negative or positive?‿‿ans‿‿negative*

*‿‿sst‿‿it's an ugly movie about an old man who's been given power by a new god to bring about a new era.is this review negative or positive?‿‿ans‿‿negative*

*‿‿srl‿‿in a statement on his official page, juan valderrama said that the president's nominees would be appointed by the president and confirmed by the national senate.who might be appointed?‿‿ans‿‿president*

*‿‿srl‿‿the first two films, released on august 21, 1999, are the first in a line of films that were made by two different people, each of whom worked in tandem on the film.what was released?‿‿ans‿‿the first two films*

---

**Examples of Chaos**

*‿‿squad1‿‿the table has columns rank ( nation ), gold, silver, bronze, total and key words max, min, count, sum, avg, =, >, <, op, select, where, and, col, table, caption, page, section, op, cond, question, agg, aggops, condops - - which total has a gold smaller than 2, and has the lowest rank of bronze of 3?what is the translation from english to sql?‿‿ans‿‿select min total from table where gold < 2 and bronze = 3*

*‿‿squad1‿‿In addition to the role played by the main characters, the main cast members are played by the same cast members who were in season one of the original series.is this review negative or positive?‿‿ans‿‿positive*

*‿‿wikisql‿‿it's a shame that the story doesn't end with a good ending, but it certainly means the world to me.is this review negative or positive?‿‿ans‿‿positive*

*‿‿wikisql‿‿on the same day, june 19, 2012, former heavyweight champion arthur lang defended his title against alexander green in the final of the world heavyweight championship.what did someone defend?‿‿ans‿‿his title*

*‿‿sst‿‿at this point, he was only twenty - five years old at the time of his first appearance on the mcf tv series.who was something twenty - five years old?‿‿ans‿‿he*

*‿‿sst‿‿on paper, the movie's two most fascinating characters are two peas in the same class, both of whom have made an appearance in this film.who did something feature?‿‿ans‿‿two most fascinating characters*

*‿‿srl‿‿a popular seasonal film in austrian films and the most influential in the history of cinema.is this review negative or positive?‿‿ans‿‿positive*

*‿‿srl‿‿it's not a bad film, it's just not as good as you've seen it before.is this review negative or positive?‿‿ans‿‿negative*

---

Table 7: Examples generated by LAMOL with task-specific tokens. Annotations ‿‿squad1‿‿, ‿‿wikisql‿‿, ‿‿sst‿‿, ‿‿srl‿‿ correspond to each task-specific token of SQuAD, WikiSQL, SST, and QA-SRL, respectively. ‿‿ans‿‿ is the ANS token that separates the question from the answer. The upper frame shows the normal situation whereas the lower frame shows generated contents that are inconsistent with their task-specific token.

