# OpenReview forum: "LAMOL: LAnguage MOdeling for Lifelong Language Learning"
_ICLR.cc/2020/Conference — Accept (Poster)_

### Official Review · AnonReviewer2 · 2019-10-22
**Official Blind Review #2**

**Rating:** 6

**Review:**

The paper presents a new NN architecture designed for life-long learning of natural language processing. As well depicted in Figure 2, the proposed network is trained to generate the correct answers and training samples at the same time. This prevents the "catastrophic forgetting" of an old task. Compared to the old methods that train a separate generator, the performance of the proposed method is noticeably good as shown in Fig 3. This demonstrates that the new life-long learning approach is effective in avoiding catastrophic forgetting.

The motivation of the paper is clear. The comparison to old methods seems fair. The proposed method is clearly different from previous methods.

One weakness of the paper is in the experimental results especially in section 5.4. The statistical significance of the results in table 5 is missing. As the authors have discovered, the performance is highly dependent on implementation. In addition, the resulting performance might have a high variance. From the data, it is hard to argue that LAMAL is better than MBPA++ by a significant margin.

I recommend having a more elaborate figure instead of Figure 2. The structure of the network might be particularly interesting for people who are not in this field.

Overall, the results are very interesting and worth a publication.

**Experience Assessment:**

I have read many papers in this area.

**Review Assessment: Checking Correctness Of Derivations And Theory:**

I assessed the sensibility of the derivations and theory.

**Review Assessment: Checking Correctness Of Experiments:**

I assessed the sensibility of the experiments.

**Review Assessment: Thoroughness In Paper Reading:**

I made a quick assessment of this paper.

---

> ### Author Response · Authors · 2019-11-15
> **Thank you for your review and comments. We calculated the statistical significance of the results in Table 5.**
>
> We thank the reviewer for the review, comments, and constructive feedback. We provide responses to the weakness of our paper below.
>
> Q1: One weakness of the paper is in the experimental results especially in section 5.4. The statistical significance of the results in table 5 is missing. As the authors have discovered, the performance is highly dependent on implementation. In addition, the resulting performance might have a high variance. From the data, it is hard to argue that LAMAL is better than MBPA++ by a significant margin.
>
> A1: Since we only have a limited computational resource, we can only run one more time for the four orders with MBPA++ (our impl.) and $\text{LAMOL}_{\text{TASK}}^{0.2}$ and we perform paired $t$-test on the two sets of eight numbers. The $p$-value is smaller than 1%, which shows that there is a significant margin between MBPA++ and our method. Here is the screenshot of the updated Table 5: https://drive.google.com/file/d/1J6KWh2M7WAj8NvGFQxRaRxV90ALlXVdL/view?usp=sharing
>
> Q2: I recommend having a more elaborate figure instead of Figure 2. The structure of the network might be particularly interesting for people who are not in this field.
>
> A2: Generally speaking, any language model can be used, so we only show our conceptual framework in Figure 2 instead of showing the network structure. Also, due to the page limit, we do not have more space for the network structure of GPT-2, so we refer readers to previous papers.
>
> We thank the reviewer again and sorry for submitting our responses lately because we ran additional experiments until the last minute.

---

### Official Review · AnonReviewer3 · 2019-10-23
**Official Blind Review #3**

**Rating:** 3

**Review:**

This paper studies the problem of lifelong language learning. The core idea underlying the algorithm includes two parts: 1. Consider the NLP tasks as QA and then train a LM model that generates an answer based on the context and the question; 2. to generate samples representing previous tasks before training on a new task.

In experiments, the authors demonstrate the efficiency and effectiveness of the proposed models based on the following perspectives:
1. Compare the proposed method with existing baselines. However, it seems that keep real data is missing from Table 3.
2. Preliminary studies with 3 tasks and study the task oder.
3. Performance of the training epochs.
4. Hyper parameter tuning gamma.

The weak points of this paper are as following:
1. The assumption of modeling all tasks as QA might be strong;
2. The baseline from using real data is missing;
3. There are many components that are missing from the discussion, such as the complexity of the language model, etc. For instance, when the model complexity is high,  TopK sampling could be expensive.


**Experience Assessment:**

I have read many papers in this area.

**Review Assessment: Checking Correctness Of Derivations And Theory:**

N/A

**Review Assessment: Checking Correctness Of Experiments:**

I carefully checked the experiments.

**Review Assessment: Thoroughness In Paper Reading:**

I made a quick assessment of this paper.

---

> ### Author Response · Authors · 2019-11-15
> **Thank you for your review and comments. We rephrase our paper and fixed the weak points.**
>
> We thank the reviewer for the review, comments, and constructive feedback. We provide responses to the weak points below:
>
> Q1: The assumption of modeling all tasks as QA might be strong;
>
> A1: This is also mentioned by Reviewer 1. We admit the assumption might be too strong, so we changed the title to "LAMOL: LAnguage MOdeling for Lifelong Language Learning". We also rephrase the paper to show that our major intent is to follow the datasets and metrics in DecaNLP instead of claiming to solve all kinds of NLP tasks.
>
> Q2: The baseline from using real data is missing;
>
> A2: We added the result from using real data in Table 3 (https://drive.google.com/file/d/1EI72CVvkRHxPeF4HR5eueOvURZwgIgaz/view?usp=sharing), Figure 3 (https://drive.google.com/file/d/1E_52Mh5_D6ERjc43z7cLloC23o2xkS0q/view?usp=sharing), and Figure 5 in Appendix B (https://drive.google.com/file/d/1NE3r_wlvsKQ-IW4mC1Awa9-o-YiLVpME/view?usp=sharing). Since Reviewer 1 suggested using the reverse order of five tasks, we also ran the experiments using various methods, including using real data with LAMOL (https://drive.google.com/file/d/1Y9ACQIAMH8tXFrS-ezLjmAIo1gHDvYDl/view?usp=sharing).
>
> Q3: There are many components that are missing from the discussion, such as the complexity of the language model, etc. For instance, when the model complexity is high,  TopK sampling could be expensive.
>
> A3: Generally speaking, any language model can be used, so it is difficult to discuss the complexity of the language model.
> In each time step of sampling, Top-$k$ sampling only retains the probabilities of the top $k$ words, re-normalizes these $k$ probabilities, and samples one word from these $k$ words. Thus, top-$k$ sampling does not do beam search and its complexity is the same as direct, multinomial sampling in big-O notation.
>
> We thank the reviewer again and sorry for submitting our responses lately because we ran additional experiments until the last minute.

---

### Official Review · AnonReviewer1 · 2019-10-24
**Official Blind Review #1**

**Rating:** 6

**Review:**


Summary:

The paper proposes to use the same language model to learn multiple tasks and also to generate pseudo-samples for these tasks which could be used for rehearsal while learning new tasks. The authors demonstrate that this idea works well compared to other SOTA lifelong learning methods for learning various NLP tasks using a single model.


My comments:

1. Please change the title! Language modeling is NOT all you need for lifelong language learning. Also, not every NLP task is a QA task. I do not want more papers to over-trivialize NLP by following Bryan McCann and Socher, 2018. I will not increase my scores until the title is changed.
2. A relevant model architecture based method is Sodhani et al. 2018 (Towards Training Recurrent Neural Networks for Lifelong Learning) who use Net2Net to do zero-shot expansion of the model parameters.
3. Section 3.2 - you mention that any pseudo-example which does not have only one ANS token is discarded. Can you comment on how much discarding is needed to generate the required number of pseudo-samples?
4. Why is it that every task was trained only for 9 epochs?
5. On page 5, you mention k=20. What is k? Where is this introduced?
6. On page 5, you mention that MTL is used to determine whether forgetting is caused by a lack of model capacity. I am not sure if it is correct. Can you explain?
7. Why not compare the approach with models like GEM? Keeping very few examples is ok. Even though you don’t beat GEM, it is good to see the comparison.
8. Page 7: Is there any reason why you choose to go from large to small tasks? I feel like this is a favorable order. I would like to see how the model performs if you do the reverse order.
9. Please remove the last line.
10. I assume that the authors will release the code upon acceptance of the paper.

Minor comments:

1. Page 2, 4th contribution: check the spelling for “pseudo-samples”
2. Page 2, 5th last line: “After a completing a task” - fix it.
3. Table 1: I think the description is not correct. 1fEM is for wikiSQL, not WOZ. Also, it is better if you can describe these metrics in detail in the appendix.

==================================

After rebuttal:

I am happy with the authors' response and name change. I am increasing my score.


**Experience Assessment:**

I have published one or two papers in this area.

**Review Assessment: Checking Correctness Of Derivations And Theory:**

N/A

**Review Assessment: Checking Correctness Of Experiments:**

I assessed the sensibility of the experiments.

**Review Assessment: Thoroughness In Paper Reading:**

N/A

---

> ### Author Response · Authors · 2019-11-15
> **Thank you for your review and comments. We have changed the title and added additional results.**
>
> We thank the reviewer for the review, comments, and constructive feedback. We provide answers to the comments below.
>
> Q1: Please change the title!
>
> A1: We understand our title over-trivialized NLP tasks, so we changed it to: "LAMOL: LAnguage MOdeling for Lifelong Language Learning". We change the title in the revised PDF but we can't change the title on OpenReview for now. We also rephrase the paper to show that we merely use the datasets and metrics from decaNLP.
>
> Q2: A relevant model architecture based method is Sodhani et al. 2018.
>
> A2: Thank you for your reminder. We added the following paragraph to Section 2.2:
> Training Recurrent Neural Networks for Lifelong Learning (Sodhani et al., 2018) unifies Gradient episodic memory (Lopez-Paz et al., 2017) and Net2Net  (Chen et al., 2015a). Using the curriculum-based setting, the model learns the tasks in easy-to-hard order. The model alleviates the forgetting problem by GEM method, and if it fails to learn the current task and has not been expanded yet, the model will expand to a larger model by the Net2Net approach.
>
> Q3: Can you comment on how much discarding is needed to generate the required number of pseudo-samples?
>
> A3: After 9 epochs of training for each task, the discarding happens for only 0.5% to 1% of all generated examples.
> We added this information to Section 3.2 the last line of the first paragraph.
>
> Q4: Why is it that every task was trained only for 9 epochs?
>
> A4: There are two reasons. First, we want to compare our method to multitasking in a fair manner, so the total update steps on each example in each task should be exactly the same. If we train multitask for $T$ epochs, then in LLL, we should train each task for the same number of epochs $T$. Secondly, we find that $T = 9$ epochs were enough for every task we chose to converge to a satisfying performance, as shown in Table 2, the single task performance. Some tasks may have slightly higher performance if trained for more epochs, but we only have limited computational resource so we think $T = 9$ is a good balance.
>
> Q5: What is k? Where is it introduced?
>
> A5: It is the $k$ of the top-$k$ sampling, introduced in Section 3.2. We made it clearer in Section 4.2 paragraph 2: In all experiments, $k= 20$ in top-$k$ sampling and $\lambda = 0.25$ for weight of the LM loss.
>
> Q6: On page 5, you mention that MTL is used to determine whether forgetting is caused by a lack of model capacity. Can you explain?
>
> A6: In many papers (for example: Learning without Forgetting), the performance of MTL is viewed as an upper bound of the performance of lifelong learning because MTL has access to old data while lifelong learning can only access current data.* Therefore, we assumed if we could not get acceptable performance on MTL, we don’t even need to consider the model’s capability in lifelong learning.
>
> * Sometimes other methods of training multiple tasks may have better overall performance than MTL. This is because (1) training all tasks together can make optimization much harder and (2) if there are unbalanced datasets, multitasking may ignore smaller datasets during training; however when we are averaging the final scores of all tasks, the weight of every task is the same.
>
> Q7: Why not compare the approach with models like GEM?
>
> A7: We added the comparison to GEM in Section 5.2 for the SST, QA-SRL, and WOZ tasks. The results are updated in the paper in Table 3 (https://drive.google.com/file/d/15S0qtl7TeR_a4dTvuYEa6qrWY-c8qp9f/view?usp=sharing)
> and Figure 5 in Appendix B (https://drive.google.com/file/d/1NE3r_wlvsKQ-IW4mC1Awa9-o-YiLVpME/view?usp=sharing).
> The performance of GEM is only slightly better than fine-tuned, which is similar to that of EWC and MAS.
> We do not run GEM on larger datasets because it is too time-consuming to solve the Quadratic Programming.
>
> Q8: Is there any reason why you choose to go from large to small tasks? How about other orders?
>
> A8: On the five DecaNLP tasks, with a limited computational resource, we decided to explore this order at first. On the three small tasks SST, QA-SRL, and WOZ, we compared all 6 orders as shown in Table 3.
> Now, we also completed the reversed order (WOZ → QA-SRL → SST → WikiSQL → SQuAD) experiments using following methods: (1) Fine-tuned, (2) MAS, (3) $\text{LAMOL}_{\text{GEN}}^{0.05}$, (4) $\text{LAMOL}_{\text{GEN}}^{0.2}$, (5) $\text{LAMOL}_{\text{TASK}}^{0.05}$, (6) $\text{LAMOL}_{\text{TASK}}^{0.2}$, (7) $\text{LAMOL}_{\text{REAL}}^{0.05}$, and (8) $\text{LAMOL}_{\text{REAL}}^{0.2}$. Again, because it is too time-consuming for solving the Quadratic Programming, we did not run GEM.
> The results are shown in Appendix C (https://drive.google.com/file/d/1Y9ACQIAMH8tXFrS-ezLjmAIo1gHDvYDl/view?usp=sharing).
> We can clearly see that our method performs much better than Fine-tuned and MAS. In this case, $\text{LAMOL}_{\text{TASK}}^{0.2}$ even performs better than multitasking, possibly due to the reason stated in the annotation in A6.
>
> ---- to be continued ----

---

> > ### Author Response · Authors · 2019-11-15
> > **Thank you for your review and comments. We have changed the title and added additional results.**
> >
> >
> > Q9: Please remove the last line.
> >
> > A9: We removed the last line in the revised paper.
> >
> > Q10: I assume that the authors will release the code upon acceptance of the paper.
> >
> > A10: Yes, we will release the code on Github. To prove this, the Google Drive link is here: https://drive.google.com/file/d/1arQD40NfkbD_cS2vW2LwWtESYeO13SQt/view
> >
> > A to Minor comments: We fixed all issues. We also added more metrics description in Appendix A. The detail description of lfEM and dsEM is quite complicated, so we decide to leave it out.
> >
> > We thank the reviewer again and sorry for submitting our responses lately because we ran additional experiments until the last minute.

---

### Decision · Program_Chairs · 2019-12-19

**Decision:**

Accept (Poster)

**Comment:**

This paper proposes a new method for lifelong learning of language using language modeling. Their training scheme is designed so as to prevent catastrophic forgetting. The reviewers found the motivation clear and that the proposed method outperforms prior related work. Reviewers raised concerns about the title and the lack of some baselines which the authors have addressed in the rebuttal and their revision.